# Modeling a Dual-Mode Controller Design for a Quasi-Resonant Flyback Converter

**Ching-Chun Chuang** [1,*] , **Chih-Chiang Hua** [2], **Chong-Yu Huang** [2] **and Li-Kai Jhou** [2]

1   Bachelor Program in Interdisciplinary Studie, National Yunlin University of Science and Technology, 123 University Road, Section 3, Douliou, Yunlin 64002, Taiwan
2   Department of Electrical Engineering, National Yunlin University of Science and Technology, 123 University Road, Section 3, Douliou, Yunlin 64002, Taiwan; huacc@yuntech.edu.tw (C.-C.H.); zhenchiman@gmail.com (C.-Y.H.); triumph852963@gmail.com (L.-K.J.)
*   Correspondence: austincc@yuntech.edu.tw; Tel.: +886-5-534-2601 (ext. 4285)

**Abstract:** The proposed system can overcome the disadvantage of a high peak current in quasi-resonant fly-back (QRF) converters when operated under heavy load conditions. The operating mode and control scheme of a QRF converter with dual-mode control were established and analyzed. The dual-mode control scheme not only enabled a valley-switching detection technique that satisfied the zero-voltage switching condition but also provided a constant frequency mechanism to reduce the conduction loss in QRF converters when operated in a continuous conduction mode and under heavy load conditions. The small-signal equivalent circuit model of QRF converter circuits was constructed using an average approximation method. The technological advancement of a QRF converter with a dual-mode controller was presented in this study. The circuit simulation result of the proposed QRF converter with a mix control scheme proved that the derived circuit component parameters meet the requirements of the converter.

**Keywords:** quasi-resonant fly-back converter; valley switching detection technique; dual-mode control

## 1. Introduction

To improve the efficiency of an adapter operated under light load conditions, the adapter should be operated under discontinuous conduction mode (DCM), primarily due to the low current, zero-voltage-switching characteristic of the main power switch, and the zero-current switching characteristic of the output diode [1–5]. This can improve the efficiency of the power stage circuit and the power density characteristics [6–9]. An analysis of the active-clamp fly-back converter was proposed in [10]. In this research, a symmetrical control was used to drive the power switche, and the efficiency improved by 4% compared to the traditional RCD configuration. A fly-back converter with variable frequency control not only provides valley-switching but also reduces electromagnetic interference. Quasi-resonant fly-back (QR) converters operated under low and medium electrical loads are considered satisfactory due to their small size, low cost, and high efficiency. A QRF converter with a discontinuous conduction switching action should be operated in continuous conduction mode (CCM). A QRF converter has a high peak current and high switching loss, which reduces the efficiency of the converter when the main switch is turned off under a heavy load condition [11–15]. This study used an adaptive valley-switching circuit under a light load condition, and one strategy circuit using a constant limit switch to turn off the timing when the heavy load caused the input current in the fly-back converter to enter the CCM. Moreover, the conduction loss in a fly-back converter is reduced when a heavy load is applied [16–21]. In recent years, a strategy for mitigating transients of balanced and unbalanced system has been proposed, as presented in [22]. The supplemental strategy

of Set point automatic adjustment with correction enabled (SPAACE) has been used to enhance its performance and extend its application to more generic electrical systems. The proposed algorithm had the following characteristics:

(1). A quadratic prediction strategy for SPAACE to increase prediction accuracy.
(2). A dead-zone band supplemental strategy for SPAACE to improve set point tracking.
(3). An approach to implement SPAACE in unbalanced systems.

In this paper, a dual-mode fly-back converter is proposed. This converter has the advantages of high efficiency when a light load is applied, and low current stress due to the power switch when a heavy load is applied for low-power applications. In continuous and discontinuous conduction modes, both fixed-off time control and valley-switching control were applied to the converter [23]. This control scheme circuit will be vital for future QRF converters. However, the control method and stability of the converter can be explained using a control design unit. Therefore, this paper describes the operation of a dual-mode control QRF converter and analyzes its stability under dual-mode control.

## 2. QRF Converter with Dual-Mode Control

In order to illustrate the principle of a QRF converter with Dual-Mode Control, a fly-back converter circuit is shown in Figure 1. The fly-back transformer consists of a magnetizing inductor $L_m$, a leakage inductor $L_{lk}$, and a primary side inductor $L_p$. The turn ratio is n. The other elements are the power switch $Q_1$, diode $D_1$, and drain-source capacitor $C_{oss}$ in the QRF converter. In this section, the operation mode and control strategy are described. The design consideration of passive components and a small-signal model are derived.

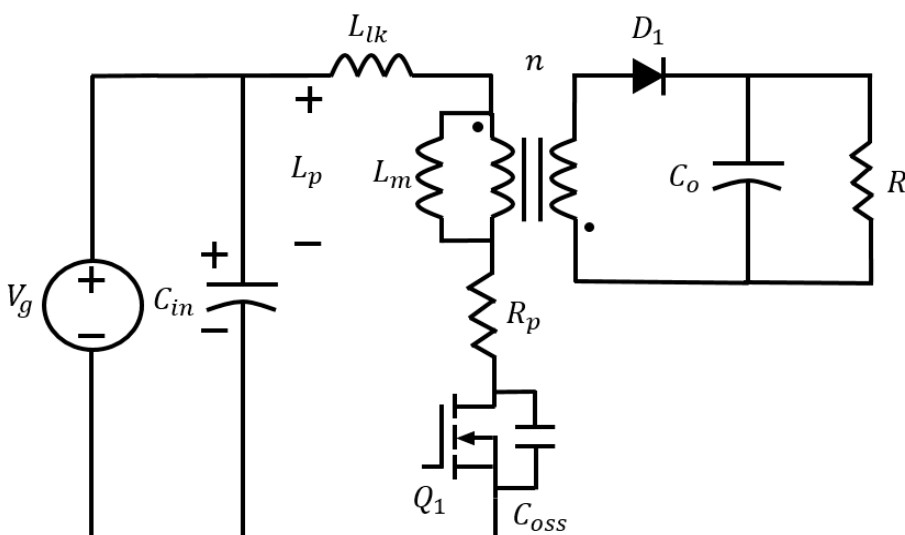

**Figure 1.** Fly-back converter circuit.

### 2.1. Single-Switch Flyback Converter

QRF converters operated in DCM are designed such that they lower the switching loss in the circuit. Because of the resonance, $L_{lk}$, $C_{oss}$, and a voltage spike across the drain and source of $Q_1$ were observed. $L_p$ is the primary inductance that comprises $L_m$ and $L_{lk}$. The primary reason for the generation of leakage inductance is that energy stored in the primary side of the converter cannot couple to the secondary-side. This energy must be transferred. Hence, when the switch turns off, the energy of $L_{lk}$ causes a voltage spike in the drain-source voltage. $C_{oss}$ releases the energy of $L_{lk}$ through the discharging path if the switch is turned on, and a voltage spike is induced. Figure 2a,b display the voltage of the switches for the DCM fly-back converter and QRF converter respectively.

The analysis method of each operation mode of a QRF converter is discussed herein. The relationship between the lowest frequency and the primary inductance is also presented in the following section.

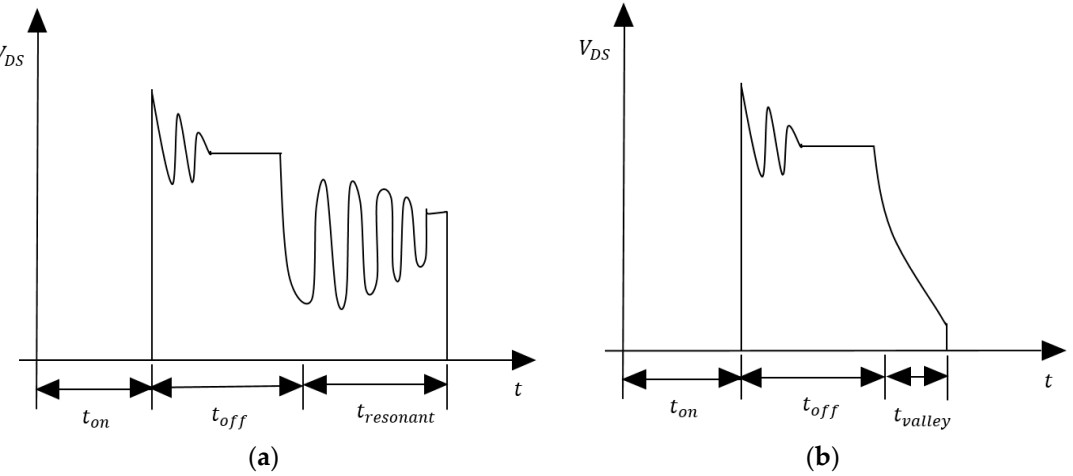

**Figure 2.** Drain-source voltage of the power switch used in a discontinuous conduction mode (DCM) fly-back converter and quasi-resonant fly-back (QRF) converter with a discontinuous conduction mode: (**a**) drain-source voltage of the power switch in the DCM fly-back mode and (**b**) drain-source voltage of the power switch in the QRF operation mode.

When $Q_1$ conducts, $V_g$ begins to charge to $L_p$. The current in the inductor can be expressed as follows:

$$I_{Lp}(t) = \frac{V_g}{L_p}t \tag{1}$$

At $T_{on}$, the peak current in the inductor $I_{Lp(pk)}$ can be represented as follows:

$$I_{Lp(pk)} = \frac{V_g}{L_p}T_{on} \tag{2}$$

When $Q_1$ is turned off, the energy stored in $L_{lk}$ cannot be transferred to the secondary side. Hence, $C_{oss}$ begins to resonate with $L_{lk}$. The voltage spike on the power switch ($V_{ds(\max)}$) is generated as follows:

$$V_{ds(max)} = V_g + nV_o + I_{Lp(pk)}\sqrt{\frac{L_{lk}}{C_{oss}}} \tag{3}$$

When the resonant energy stored in $L_{lk}$ and $C_{oss}$ reaches zero during the degaussing, the drain source voltage ($V_{ds}$) drops to the constant voltage until the magnetizing inductor releases energy. $V_{ds}$ can be expressed as follows:

$$V_{ds} = V_g + nV_o \tag{4}$$

The primary current during $T_{off}$ can be derived as follows:

$$I_{Lp(pk)} = \frac{nV_o}{L_p}T_{off} \tag{5}$$

In the steady state, when the secondary side does not transmit current through the transformer, the primary side behaves as an RLC resonant circuit:

$$V_{ds}(t) = V_g + nV_oe^{-(\frac{Rp}{2L_p})t}\cos\left(2\pi\frac{1}{2\pi\sqrt{L_pC_{oss}}}t\right) \tag{6}$$

where $R_p$ is the total resistance of a primary side path of transformer.

The fly-back converter enters the resonant state when the power switch of the QRF converter is turned off. The first valley switching time at a value of $T_{valley}$ can be obtained using the following representation of $T_{valley}$:

$$\cos\left(2\pi f_w T_{valley}\right) = -1 \tag{7}$$

and

$$T_{valley} = \frac{1}{2f_w} = \pi \sqrt{L_p C_{oss}}. \tag{8}$$

The underdamped response case of the fly-back converter affects the drain-source voltage until the power switch conducts power again during $T_{resonant}$. The drain source voltage of the QRF converter produces an underdamped oscillation until the power switch begins to turn on at a $T_{resonant}$ value of $T_{valley}$. Thus, the switch cycle ($T_{sw}$) of the fly-back converter can be determined using the following equation:

$$T_{sw} = T_{on} + T_{off} + T_{resonant} \tag{9}$$

and

$$T_{sw} = \frac{L_p}{V_g} I_{Lp(pk)} + \frac{L_p}{nV_o} I_{Lp(pk)} + nV_o e^{-at} \cos(2\pi f_w T_{resonant}). \tag{10}$$

The switching frequency formula at this mode can be estimated as follows:

$$f_{sw} = \frac{nV_g V_o}{L_p I_{Lp(pk)} nV_o + L_p V_g I_{Lp(pk)} + nV_g V_o \frac{\pi}{2} \sqrt{L_p C_{oss}}}. \tag{11}$$

Then, the switching cycle $T_{sw}$ of the QRF converter can be obtained as follows:

$$T_{sw} = T_{on} + T_{off} + T_{valley} \tag{12}$$

and

$$T_{sw} = \frac{L_p}{V_g} I_{Lp(pk)} + \frac{L_p}{nV_o} I_{Lp(pk)} + \pi \sqrt{L_p C_{oss}}. \tag{13}$$

The final formula of the switching frequency at this mode can be estimated as follows:

$$f_{sw} = \frac{nV_g V_o}{L_p I_{Lp(pk)} nV_o + L_p V_g I_{Lp(pk)} + nV_g V_o \pi \sqrt{L_p C_{oss}}}. \tag{14}$$

In terms of the power stage design, the inductor is the primary component for input power determination. The input power of the fly-back converter ($P_{in}$) can be estimated using the following equation:

$$P_{in} = \frac{P_o}{\eta} = \frac{1}{2} L_p \left(I_{Lp(pk)}\right)^2 f_w. \tag{15}$$

The switching frequency formula can be rearranged as follows:

$$T_{sw} = \frac{1}{f_{sw}} = \frac{L_p \left(I_{Lp(pk)}\right)^2 \eta}{2P_o}. \tag{16}$$

By substituting $T_{sw}$ into (16), the following representation of $I_{Lp(pk)}$ is obtained:

$$I_{Lp(pk)} = \frac{\left(\frac{nV_o L_p + V_g L_p}{nV_g V_o}\right)\left(\frac{2P_o}{L_p \eta}\right) \pm \sqrt{\left(\frac{nV_o L_p + V_g L_p}{nV_g V_o}\right)^2 + 4\left(\pi \sqrt{L_p C_{oss}} \frac{2P_o}{L_p \eta}\right)}}{2}. \tag{17}$$

Finally, the value of $f_{sw}$ can be found by substituting (17) into (14). The relationship of $f_{sw}$ with the input voltage and output load can be expressed as follows:

$$f_{sw} = \frac{4P_o}{\eta L_p \left[ \left( \frac{nV_oL_p + V_gL_P}{nV_gV_o} \right) \left( \frac{2P_o}{L_p\eta} \right) \pm \sqrt{\left( \frac{nV_oL_p + V_gL_P}{nV_gV_o} \right)^2 + 4\left( \pi \sqrt{L_pC_{oss}} \frac{2P_o}{L_p\eta} \right)} \right]}. \tag{18}$$

The minimum switching frequency ($f_{sw(\min)}$) is obtained at $P_{o(\max)}$ and $V_{in(\min)}$ based on the above formula. To confirm the relation between $L_p$ and $f_{sw(\min)}$, the minimum inductance condition of $L_p$ can be designed for the inequality condition as follows:

$$L_p \leq \frac{4P_{o(\max)}}{\eta f_{sw(\min)} \left[ \left( \frac{nV_oL_p + V_gL_P}{nV_gV_o} \right) \left( \frac{2P_{o(\max)}}{L_p\eta} \right) \pm \sqrt{\left( \frac{nV_oL_p + V_gL_P}{nV_gV_o} \right)^2 + 4\left( \pi \sqrt{L_pC_{oss}} \frac{2P_{o(\max)}}{L_p\eta} \right)} \right]}. \tag{19}$$

The QRF converter operates in discontinuous conduction mode. Moreover, the switching frequency changes due to variations in the input voltage (Vg), output voltage (Vo), and output power (Po). Thus, the minimum limit condition of the switching frequency must be considered in the proposed circuit. Figure 3 illustrates the curve of the relationship between $f_{sw}$ and $V_g$.

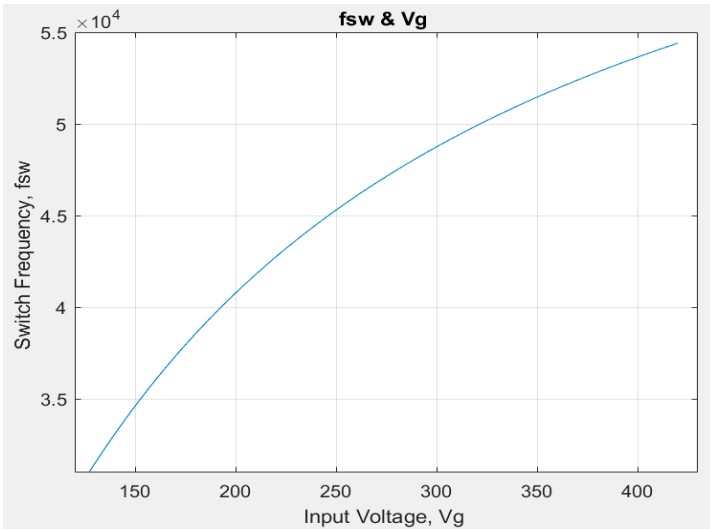

**Figure 3.** Curve of the relationship between $f_{sw}$ and $V_g$.

### 2.2. Small-Signal AC Model

In steady state, the output voltage of a DMFBC is constant. On the basis of the state average method [24], a small-signal equation of DMFBC was obtained. The small-signal equation can be represented as follows:

$$L\frac{d\hat{i}(t)}{dt} = D\hat{v}_g(t) - D'\frac{\hat{v}_o}{n} + \left( V_g + \frac{V}{n} - IR_{on} \right)\hat{d}(t) - DR_{on}\hat{i}(t) \tag{20}$$

and

$$C\frac{d\hat{v}_o(t)}{dt} = \frac{D'\hat{i}(t)}{n} - \frac{\hat{v}(t)}{R} - \frac{I\hat{d}(t)}{n} \tag{21}$$

and

$$\hat{i}_g(t) = D\hat{i}(t) + I\hat{d}(t). \tag{22}$$

The hats sign represents AC change. The relationship between stability and the equivalent series resistance of the output capacitor of a dual-mode fly-back converter (DMFBC) should be considered in greater detail, because the left-plane of the system circuit has a zero point that is produced by the output capacitor ESR which affects system stability. Moreover, the analysis technique of a conventional linear circuit was used in this proposed fly-back converter and the transfer function of DMFBC was derived for verification. To discuss the relationship between input and output voltages, first, the duty cycle was set to zero ($\hat{d}(t) = 0$). Then, the open-loop voltage transfer function of the output to the input $G_{v_o v_g}$ was derived. The input voltage ($\hat{v}_g(t) = 0$) was set and derived, and the voltage transfer function of the control to the input $G_{v_g d}$ was presented. These transfer functions describe how a control change influences input voltage. The relationships between the output voltage response and the different input voltage states were discussed. The relationship between $G_{v_o v_g}$ and $G_{v_g d}$ is the most prevalent in terms of the input voltage loop gain. Figure 4 displays a small-signal AC equivalent circuit model of a DMFBC through the aforementioned derivation.

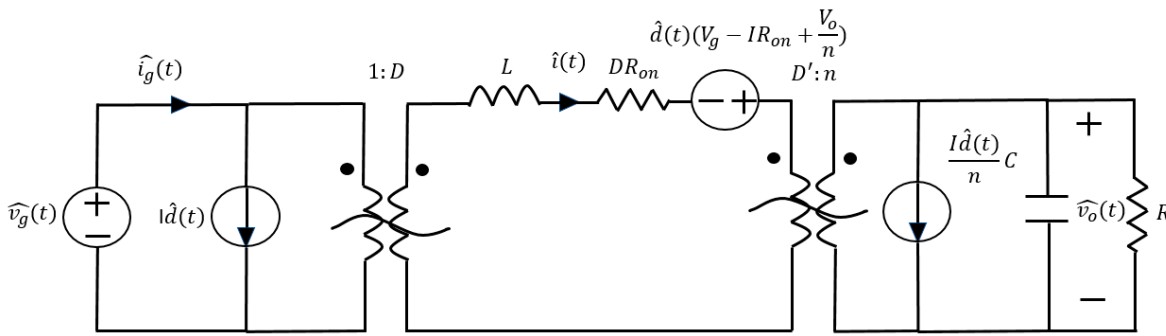

**Figure 4.** Small-signal ac equivalent circuit of the DMBFC.

Many studies have discussed the derivation of a small-signal pulse-width modulator (PWM) equivalent circuit. In addition to considering the derivative of the primary power stage of a small-stage mode that has an effective duty cycle ($d_{eff}$) to output voltage and the loop control transfer function, we derived the input voltage to output voltage loop control transfer function and the effective duty cycle to output voltage loop control transfer function. A series of mathematical formulas that describe the technical aspects of a QRF converter, the transfer functions $G_{v_o v_g}(s)$ and $G_{v_o d}(s)$, were obtained in the s-domain of QRF converter. Moreover, we analyzed and derived the transfer function of a DMFBC to reveal the relationship between a pole-zero point and an asymptotic gain model. The output to input voltage open-loop transfer function can be estimated as follows:

$$G_{v_o v_g}(s) = \left.\frac{\hat{v}_o(s)}{\hat{v}_g(s)}\right|_{\hat{d}(s)=0} = G_{do} \cdot \frac{1}{1 + \frac{s}{Qw_o} + \left(\frac{s}{w_o}\right)^2} \tag{23}$$

and

$$G_{v_o v_g}(s) = \left.\frac{\hat{v}_o(s)}{\hat{v}_g(s)}\right|_{\hat{d}(s)=0} = \left(\frac{nD}{D'}\right) \cdot \frac{1}{1 + s\frac{nL}{(D')^2} + s^2\frac{ns^2 RLC}{(D')^2}} \tag{24}$$

where

$$G_{do} = \left(\frac{nD}{D'}\right) w_o = \frac{D'}{\sqrt{nLC}} Q = \sqrt{\frac{LC}{n}} \cdot \frac{D'R}{L}.$$

$G_{do}$, $\omega_o$ and $Q$ are the analyzed keywords of the transfer function $G_{v_o v_g}(s)$ of the dual-mode control of a fly-back converter. In this mode, the physical parameter values $R$, $L$, and $C$ serve a crucial role in designing the parameters for a QRF converter. To implement these parameters, the R, L, and C values

used were respectively 3.61 Ω, 170 Ω, and 2000 μF at the heavier load. The control to input voltage transfer function can be estimated as follows:

$$G_{v_o d}(s) = \frac{\hat{v}_o(s)}{\hat{d}(s)}\Bigg|_{\hat{v}_g(s) = 0} = G_{do} \cdot \frac{1 - \frac{s}{w_z}}{1 + \frac{s}{Qw_o} + \left(\frac{s}{w_o}\right)^2} \tag{25}$$

and

$$G_{v_o d}(s) = \frac{\hat{v}_o(s)}{\hat{d}(s)}\Bigg|_{\hat{v}_g(s) = 0}$$

$$= \left(\frac{D\left(V_g - IR_{on} + \frac{V_o}{n}\right)RD' + D^2 R_{on}R}{D^2 R_{on} + (D')^2 R}\right)$$

$$\times \frac{\left[1 - s\left(\frac{\left(\frac{I}{n}\right)D^2 RL}{D^2 R_{on} + (D')^2 R}\right)\left(\frac{D^2 R_{on} + (D')^2 R}{D\left(V_g - IR_{on} + \frac{V_o}{n}\right)RD' + D^2 R_{on}R}\right)\right]}{1 + \left(\frac{D^2 L + D^2 R_{on}RC}{D^2 R_{on} + (D')^2 R}\right)s + \left(\frac{D^2 RLC}{D^2 R_{on} + (D')^2 R}\right)s^2} \tag{26}$$

and

$$G_{do} = \frac{D\left(V_g - IR_{on} + \frac{V_o}{n}\right)RD' + D^2 R_{on}R}{D^2 R_{on} + (D')^2 R} \tag{27}$$

and

$$w_z = \frac{(D^2 R_{on} + (D')^2 R)\left[D\left(V_g - IR_{on} + \frac{V_o}{n}\right)RD' + D^2 R_{on}R\right]}{\left(\frac{I}{n}\right)D^2 RL(D^2 R_{on} + (D')^2 R)} \tag{28}$$

and

$$Q = \frac{(D^2 R_{on} + (D')^2 R)}{D^2 L + D^2 R_{on}RC}\sqrt{\frac{D^2 RLC}{D^2 R_{on} + (D')^2 R}} \tag{29}$$

and

$$w_o = \sqrt{\frac{D^2 R_{on} + (D')^2 R}{D^2 RLC}}. \tag{30}$$

## 3. Controller Design

To improve the frequency response of a dual-mode fly-back converter and simultaneously implement the converter performance, an adaptive compensator for obtaining the transfer function of $G_{v_o v_g}(s)$ and $G_{v_o d}(s)$ is required. $G_{v_o d}(s)$ is an important keyword of the system loop gain in a DMFBC and can be used to determine the performances of voltage and current loop compensators. A peak current control integrated circuit (IC) L6561 with a transient mode control performance was used in the fly-back converter. The fly-back converter is equipped with an adaptive detection circuit and judge circuit, and can be operated under heavy load or light load conditions to improve the converter efficiency of the converter.

When the fly-back converter with the constant duty cycle turn off control was operated under heavy load conditions, the converter entered to the CCM, the RMS current went through the power switch, and the quick diode was reduced. In contrast, the conduction loss increased. To apply the ZCD detection function of the IC L6561 control [22], the control gate drive signal was kept off until it was turned on in the next cycle of ZCD voltage level detection.

The ZCD action function can be estimated as follows:

$$\begin{cases} V_{ZCD} > V_{ZCD_{Limit}} = 5.7V, \ GD = 0V \ (Low) \\ V_{ZCD} < V_{ZCD\_Triger} = 0.7V, \ GD = 15V \ (High) \end{cases}. \tag{31}$$

A DMFBC with a ZCD detection strategy was presented in this study. The voltage-adjustable controller, optocoupler, PWM comparator, and fly-back converter were included in the control system. The transfer function of the circuit presented in Figure 5 can be expressed as follows. Figure 6 displays

the fly-back converter and voltage regulation control system. Figure 6 presents a simplified block diagram of the dual-mode control, which is represented using dotted boxes.

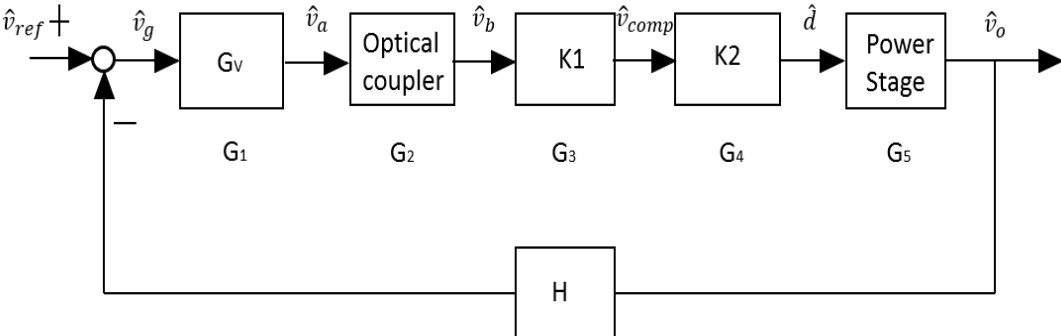

**Figure 5.** Block diagram of the voltage closed-loop control of the DMBFC.

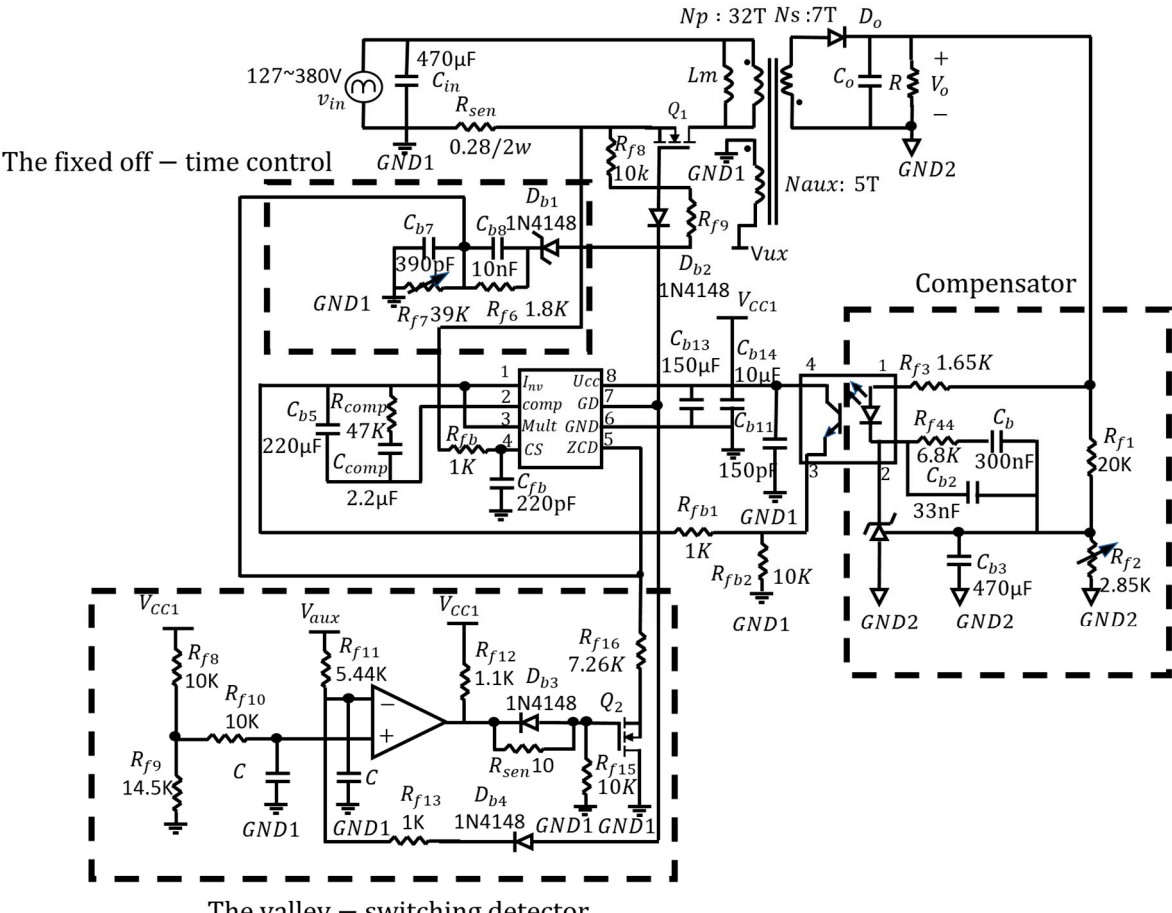

**Figure 6.** Simplified circuit diagram of dual-mode control.

$$G_1(s) = \frac{s + w_z}{R_{f4}C_{b1}s(s + w_p)} \tag{32}$$

where

$$w_z = \frac{1}{R_{f4}C_{b1}} \ (\text{rad}/sec) \quad w_P = \frac{C_{b1} + C_{b2}}{R_{f4}C_{b1}C_{b2}} \ (\text{rad}/sec)$$

and

$$G_2(s) = \frac{\hat{v}_b}{\hat{v}_a} = \frac{CTR \cdot R_{fb2}}{R_{fb3}} \tag{33}$$

and

$$G_3(s) = \frac{\hat{v}_{comp}}{\hat{v}_b} = -\left(\frac{R_{fb3}}{R_{fb1}}\right) = -\frac{\left[\dfrac{\left(\frac{1}{sC_{comp}} + R_{comp}\right)C_{b5}}{\left(\frac{1}{sC_{comp}} + R_{comp}\right) + C_{b5}}\right]}{R_{fb1}} \tag{34}$$

and

$$G_4(s) = K_2 = \frac{\hat{d}}{\hat{v}_{comp}} = \frac{1}{V_M} \tag{35}$$

and

$$G_5(s) = \frac{\hat{v}_o}{\hat{d}} = nV_g \tag{36}$$

and

$$H(s) = \frac{R_{f2}}{R_{f1} + R_{f2}} \tag{37}$$

$G_1(s)$, $G_2(s)$, $G_3(s)$, $G_4(s)$, $G_5(s)$, and $H(s)$ can be combined to obtain the open- and closed-loop voltage transfer function of a fly-back converter:

$$T_{FLY\_OPEN}(s) = G_1(s) \cdot G_2(s) \cdot G_3(s) \cdot G_4(s) \cdot G_5(s) \cdot H(s) \tag{38}$$

Moreover, the closed-loop voltage transfer function can be expressed as follows:

$$T_{FLY\_CLOSED}(s) = \frac{G_1(s) \cdot G_2(s) \cdot G_3(s) \cdot G_4(s) \cdot G_5(s)}{1 + G_1(s) \cdot G_2(s) \cdot G_3(s) \cdot G_4(s) \cdot G_5(s) \cdot H(s)} \tag{39}$$

The function characteristics of one compensator $G_V(s)$ are selected and explained by the following transfer function:

$$G_1(s) = K \frac{s + w_z}{s \cdot (s + w_p)} \tag{40}$$

Adding a zero point to the compensator reduces the phase lag and adds stability to a fly-back converter. It is expected that adding one pole point ($w_p$) suppresses the switch noise to further increase the loop gain. Furthermore, it is expected that adding one pole point to the original point of the system would increase the low frequency gain of the studied fly-back converter.

## 4. Circuit Simulations and Experimental Verifications

Computer simulations were applied to validate the controller design criteria and the formula derived from the small-signal analysis model. The actual hardware specifications are presented in Table 1. MATLAB, a simulation tool, was used to obtain the Bode plot of the converter. Moreover, the mathematical equation of the transfer function was as follows:

**Table 1.** Converters specifications.

| Symbol | Value | Symbol | Value |
| --- | --- | --- | --- |
| Vin | 127~380 Vdc | Lm | 170 $\mu H$ |
| Vo | 19V dc | Co | 2000 $\mu F/63V$ |
| Po(max) | 100 W | Dmax | 0.4 |
| Io(max) | 5.26 A | fsw(min) | 40 kHz |

The frequency response of the fly-back converter was simulated using the open-loop transfer functions $G_{v_o v_g}$ and $G_{v_o d}$ at input voltages of $127V_{dc}$ and $380V_{dc}$ and at an output voltage of $19V_{dc}$. For the compensator, a zero- and two-pole compensator (Type II) was used for the QRF converter:

$$C_{compensator}(s) = \frac{1}{R_{f4}C_{b1}} \cdot \frac{(s + \omega_z)}{s \cdot (s + \omega_p)} \tag{41}$$

This compensator can ensure closed-loop system stability, such as by increasing the low-frequency gain. For fly-back converter stability, the phase margin should be positive. As in [23], for an unconditionally stable switch mode, power conversion causes a gain margin of 6 dB and a phase margin of $45°$. The design parameters of components were derived, and a real circuit was simulated and verified to determine the performance of a QRF converter when it is operated in the AC small-signal mode. A frequency domain response was obtained by substituting the component design parameters into the voltage open-loop transfer function and the voltage closed-loop transfer function of the fly-back converter. The responses of the voltage open- and closed-loop transfer functions of the fly-back converter are presented in Figures 7 and 8, respectively. As presented in Figure 8, the fly-back converter is stable in voltage closed-loop mode, when the proposed design criterion is used because the phase margin is greater than $45°$. Moreover, the figure can be used to validate the accuracy of the derived voltage closed-loop control, because the frequency response of the simulation is consistent with the principles of Bode plots.

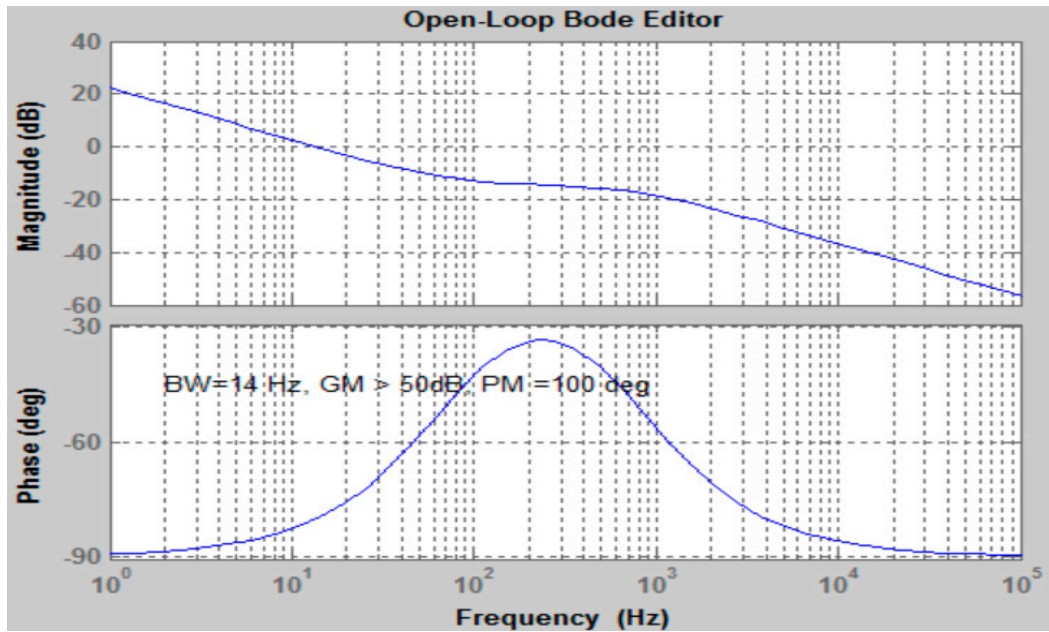

**Figure 7.** Frequency domain response of the voltage open-loop transfer function of the fly-back converter.

The principle of a valley-switching detection technique is examined in this research. The valley-switching detector of the circuit in Figure 6 comprises a switch ($Q_2$), comparator ($U_1$), and resistors ($R_{f8} - R_{f16}$). First, a voltage level ($V_{ref1}$) was applied to the valley voltage of the resonance at a half-resonance period. Then, the comparator captured the auxiliary winding feedback ($V_{aux}$) at the inverting input ($-$) pin and compared $V_{aux}$ with the non-inverter input voltages ($V_{ref1}$) in the valley-switching detector. Each comparator had a non-inverting input ($+$), an inverting input ($-$), and an output. During the duration of $t_{resonant}$ shown in Figures 2a and 6, if $V_{ref1}$ is higher than $V_{aux}$, the comparator's output is at a high level, and $Q_2$ is turned on. When the GD's output is at a high level, $Q_1$ is turned on while the ZCD pin voltage of L6561, $V_{ZCD}$ is lower than the internal voltage of L6561, $V_{ZCD\_Trigger}$ (0.7V). Figure 9 shows the valley-switching waveform of the QRF converter at a light load when the input DC voltage is $380V_{dc}$. However, when the trigger voltage signal ($V_{gs2}$) is sufficiently

higher than $V_{th}$ (the gate threshold voltage of the power switch, $V_{th}$), $Q_2$ will turn on. Also, the power switch $Q_1$ will be turned on.

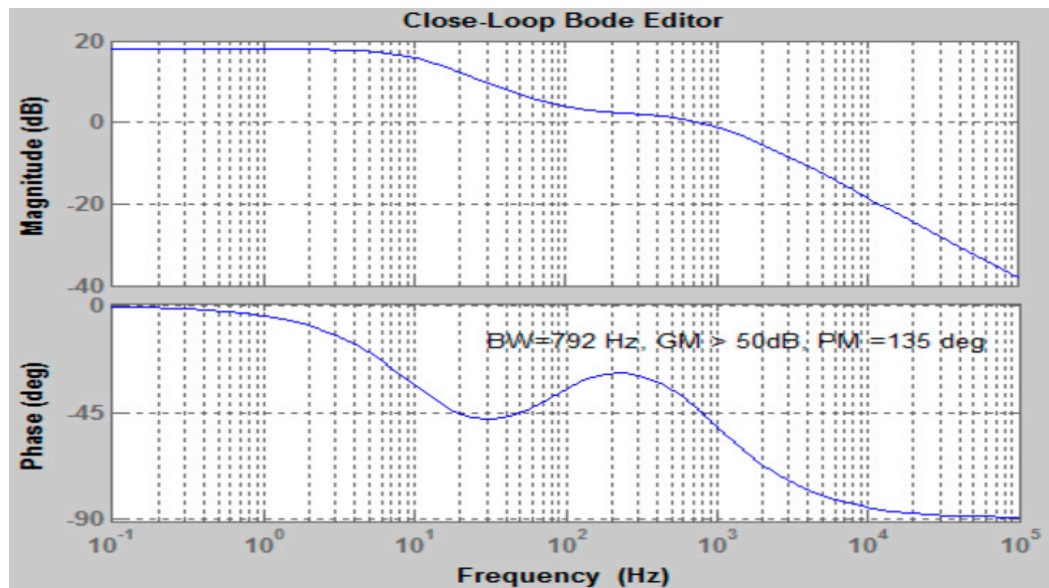

**Figure 8.** Frequency domain response of the voltage closed-loop transfer function of the fly-back converter.

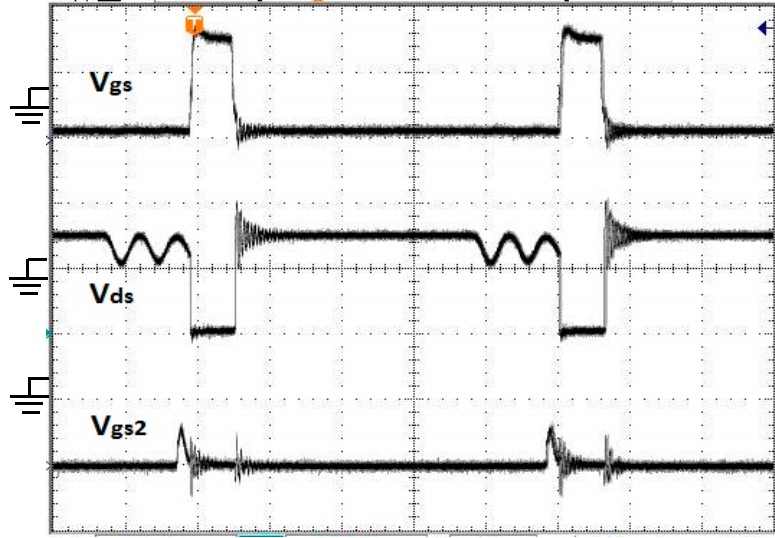

**Figure 9.** The valley-switching waveforms $V_{gs}$, $V_{ds}$, and $V_{gs2}$ at a light load when the input DC voltage is 380 $V_{dc}$. ($V_{gs}$: 10 V/div, $V_{ds}$: 10 V/div, $V_{Gs2}$: 10 V/div, time: 2 μs/div).

Figure 9 shows the experimental waveforms of the gate-to-source voltage $V_{gs}$, the drain-to-source voltage $V_{ds}$, and the gate-to-source voltage $V_{gs2}$ when the input DC voltage is 380 $V_{dc}$ at a light load. The fly-back converter is operated in DCM, and $V_{ds}$ resonates to its third valley before $Q_1$ is turned on. Figure 10b shows the waveforms of $I_d$, $I_{Q_1}$, and $V_{ds}$ when the input DC voltage is 127 $V_{dc}$ and the load current is 5.26 A. The fly-back converter is operated in CCM at the same load level. Figure 10 shows the waveforms of $I_d$, $I_Q$, and $V_{ds}$ when the input voltage is 127 $V_{dc}$ and the output power is 100 W, respectively. Figure 11 shows the waveforms of $I_d$, $I_Q$, and $V_{ds}$ when the input voltage is 380 $V_{dc}$ and the output power is 100 W, respectively.

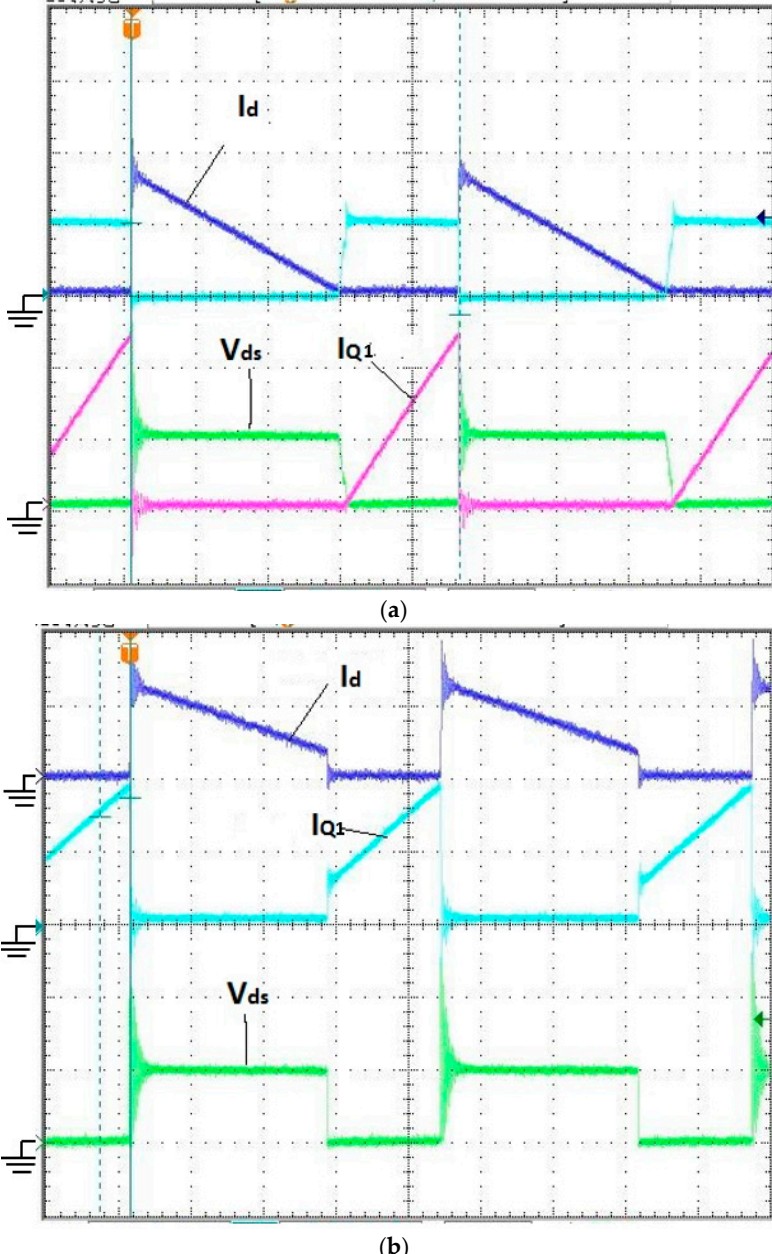

(a)

(b)

**Figure 10.** (**a**) A conventional QR-mode operation waveform, $I_o$ = 5.26 A ($I_d$: 10 A/div, $I_{Q_1}$: 2 A/div, $V_{ds}$: 200 V/div, time: 4 μ*s*/div), and (**b**) A conventional QR-mode and continuous conduction mode (CCM) operation waveform, $I_o$ = 5.26 A ($I_d$: 10 A/div, $I_{Q_1}$: 2 A/div, $V_{ds}$: 200 V/div, time: 4 μ*s*/div).

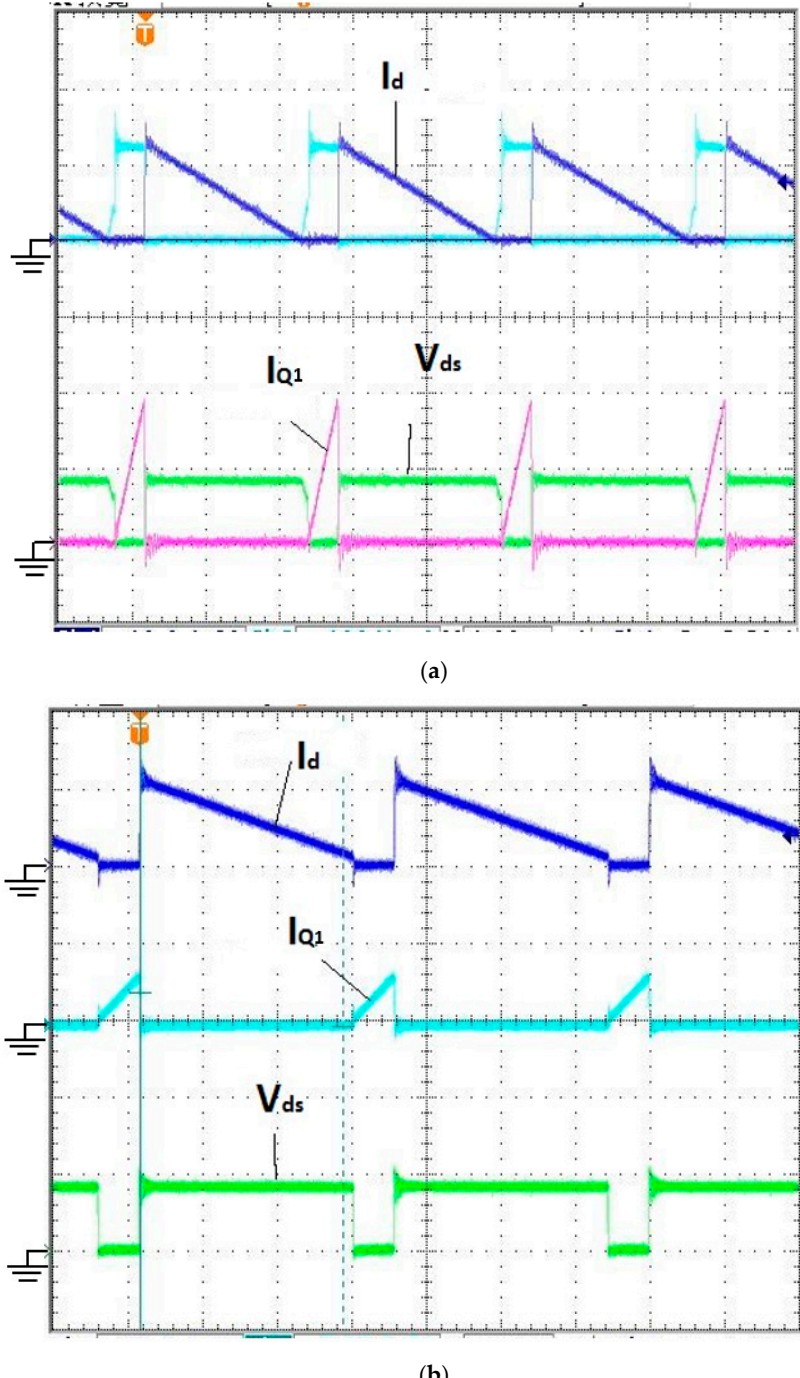

(a)

(b)

**Figure 11.** (**a**) shows a conventional QR-mode operation waveform, $I_o$ = 5.26 A ($I_d$: 10 A/div, $I_{Q_1}$: 2 A/div, $V_{ds}$: 500 V/div, time: 4 μ$s$/div), and (**b**) shows a conventional QR-mode and CCM operation waveform, $I_o$ = 5.26 A ($I_d$: 10 A/div, $I_{Q_1}$: 5 A/div, $V_{ds}$: 500 V/div, time: 4 μ$s$/div).

Table 2 presents the measured efficiencies of the proposed dual-mode controller for a QRF converter operated at an input voltage and constant output power of 127 $V_{dc}$ and 380 $V_{dc}$, respectively.

**Table 2.** Comparison with existing control mode.

| Control Mode | Vin (V) | Iin (A) | Pin (W) | Vo (V) | Io (A) | Po (W) | η (%) |
|--------------|---------|---------|---------|--------|--------|--------|-------|
| QR Control   | 127     | 0.842   | 106.68  | 18.73  | 5      | 93.65  | 87.70 |
| QR Control   | 380     | 0.281   | 106.95  | 18.81  | 5      | 94.05  | 87.90 |
| DM Control   | 127     | 0.833   | 105.41  | 18.85  | 5      | 94.25  | 89.20 |
| DM Control   | 380     | 0.277   | 105.26  | 18.90  | 5      | 94.50  | 89.70 |

The measured efficiency curves versus output powers with different input voltages are shown in Figure 12. A Voltech power analyzer (PM3000A) was used to measure the efficiency of the QRF converter. At the rated full load, the proposed conversion efficiency was about 88%.

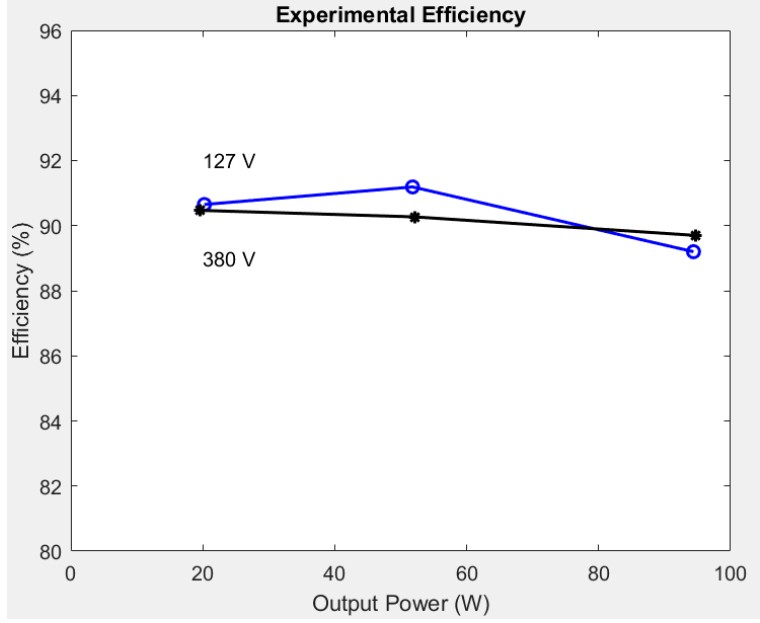

**Figure 12.** Experiment efficiencies of the dual-mode QRF converter.

## 5. Conclusions

A dual-mode control scheme for the QRF converter was used in this study. The dual-mode control scheme involved a valley-switching detection technique to satisfy the zero-voltage switching requirement. A constant frequency mechanism was proposed to reduce the conduction loss of the QRF converter when it was operated in the CCM and when a heavy load was applied. A 100 W QRF converter prototype with dual-mode control was proposed to verify the applicability of theoretical analysis. Moreover, an AC small-signal model for a QRF converter was proposed. An adaptive controller with a detection circuit that can operate under light- or heavy-load conditions was designed and used in the study. The simulation and experimental results of the proposed control scheme for the fly-back converter were demonstrated. Moreover, the accuracy of an AC small-signal model was compared and verified. An adaptive Type II compensation for a QRF converter that can operate under a heavy-load and a light-load was designed. The function and performance of a QRF circuit were evaluated to verify the practicality of the proposed dual-mode control approach.

**Author Contributions:** C.-C.C., C.-C.H., C.-Y.H. and L.-K.J. conceived and designed the prototype circuit; C.-C.C. and C.-Y.H. performed simulations and experiments, and analyzed data obtained by C.-C.C., C.-C.H., C.-Y.H. and L.-K.J.; C.-C.H. contributed equipment, materials, and analysis tools. C.-C.C., C.-C.H., C.-Y.H. and L.-K.J. wrote the paper.

**Funding:** This research was funding by the National Science Council of Taiwan under grant number MOST 107-2622-E-224-009-CC3.

**Acknowledgments:** This research was supported by the National Science Council of Taiwan under grant number MOST 107-2622-E-224-009-CC3. The equipment used for experiments was provided by the Basic Research Program, Renewable Energy Research Center, National Yunlin University of Science and Technology.

**Conflicts of Interest:** The authors have no conflict of interest to declare.

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
