# Peer review of "Modeling a Dual-Mode Controller Design for a Quasi-Resonant Flyback Converter"

_applsci, doi:10.3390/app9091860_

Reviewer 1 Report

This paper proposes a dual-mode control scheme for the QRF converter. The topic is interesting; however, the technical contribution still needs to be clarified. Moreover, there are some points that should be addressed by the authors.

•The quality of figures is not acceptable. Authors need to revise the simulation results and other figures. 

Authors described existing literature in a satisfactory way, however, most of the mentioned works introduce controllers to improve disturbance rejection performance of inverter. Authors should extend their literature review for other control methods that cover both tracking performance and disturbance rejection aspects. Reviewer suggestion would be to review and implement the works described in:

“Utilization of an active-clamp circuit to achieve soft switching in flyback converters,” IEEE.

“Predictive set point modulation to mitigate transients in lightly damped balanced and unbalanced systems,” IEEE

• The effectiveness of the proposed method under unbalanced network condition should be evaluated. The authors can add a case study and compare the performance of the system under unbalanced network condition with and without the proposed method.

Author Response

Point 1: This paper proposes a dual-mode control scheme for the QRF converter. The topic is interesting; however, the technical contribution still needs to be clarified. Moreover, there are some points that should be addressed by the authors

Response 1: Thanks a lot for taking your time and reading our work carefully as well as for giving us useful comments to improve its quality.

Point 2: The quality of figures is not acceptable. Authors need to revise the simulation results and other figures.

Response 2: Thanks a lot for reading our work carefully and minutely as well as for giving us useful comments to improve its quality. First of all, we will redraw the new Fig. 6 (Simplified circuit diagram of dual-mode control) in the revised manuscript for readability. And then the simulation result is also rearranged on page 12 of a revised manuscript.

Point 3: Authors described existing literature in a satisfactory way, however, most of the mentioned works introduce controllers to improve disturbance rejection performance of inverter. Authors should extend their literature review for other control methods that cover both tracking performance and disturbance rejection aspects. Reviewer suggestion would be to review and implement the works described in:

(1). Watson, R.; Lee, F.C.; Hua, G.C. Utilization of an active-clamp circuit to achieve soft switching in fly-back converters. IEEE Trans. Power Electron. 1996, 11, 162-169

(2). H. Ghaffarzadeh, C. Stone; and A. Mehrizi-Sani, Predictive Set Point Modulation to Mitigate Transients in Lightly Damped Balanced and Unbalanced Systems. IEEE Trans. Power Electron., vol, 32, no. 2, pp. 1041-1049, March 2017.

Response 3: First, we reviewed the literature [10] to infer a tracking performance and disturbance rejection for the control method of active-clamp converter. The author proposed that the analysis of the active-clamp fly-back converter. In the concept of the literature paper [10], fig. 2 and fig. 3 show the seven topological stages of the converter during one switching cycle and its key waveforms, respectively. A symmetrical control schemes is used for driving the power switches. Secondly, the author shown that the efficiency comparison between RCD and active-clamp configurations in Fig. 4. It is obvious that the active-clamp circuit yields an improvement of at least 4% in power stage efficiency.

See [10], for more details.

[10]. Watson, R.; Lee, F.C.; Hua, G.C. Utilization of an active-clamp circuit to achieve soft switching in fly-back converters. IEEE Trans. Power Electron. 1996, 11, 162-169.

We include the following paragraph for reviewing the control method in literature [9], as well as for clarifying the extend of performance of converter.

Thirdly, we reviewed the literature [22] (we add to the literature on the revised manuscript) to

infer a tracking performance and disturbance rejection for the control method of predictive set

point modulation in lightly damped balanced and unbalanced systems.

The author proposes an approach to implement SPAACE in unbalanced systems. The objective

of set point automatic adjustment with correction enabled (SPAACE) algorithm is mainly to

ensure that the trajectory of power system response. The supplemental strategies of SPAACE

is to enhance its performance and extend its application to more generic electric systems. A lot

of contribution of the proposed algorithm as follow:

1). A quadratic prediction strategy for SPAACE for increased prediction accuracy;

2). A dead-zone band supplemental strategy for SPAACE to improve set point tracking;

3). An approach to implement SPAACE in unbalanced systems.

We include the following paragraph for reviewing the control method in literature [22], as well

as for clarifying the extend of performance of converter

Point 4: The effectiveness of the proposed method under unbalanced network condition should be evaluated. The authors can add a case study and compare the performance of the system under unbalanced network condition with and without the proposed method.

Response 4:

We add the following paragraph and Figure 12 for clarifying the efficiency of QRF converter.

We would like to sincerely thank you for your advices and constructive comments.

Sincerely,

Ching-Chun Chuang on behalf of all the authors.

Reviewer 2 Report

Dear Authors, 

Thank you for the interesting paper. I have the following comments for the Authors: 

·       Page1, Line 42: 

·       The authors claim that are proposing “dual mode Fly-back converter”. However this has been in application for long time (see J.N. Park - A DUAL MODE FORW ARD/FLYBACK CONVERTER- IEEE Power Electronics Specialists conference 1982). Could you please explain the difference between the two concepts? Is the 

Page 2, Line 55: there is a typo in the “Conclusions are provided in section 6”. I think the authors mean to say section 5. 

Please use the equation numbers when substituting, or deriving one from the other: for example, in page 5 line 118, better to say “By substituting Tsw into (16)” as you used correctly in line 120. Consistency is important for ease of reading. 

Page 7, line 169: the passive components R, L, and are considered as having a crucial role in designing the parameters for QRF. It is important to give their specific importance in a sentence or two. 

Figure 6 is too busy with unnecessary component values and the yellow color is also a wrong choice. You can list the values in a table for readability. 

Page 11, Line 232:

“We are asked to increase …” I am not sure what you want to say here.  

Thanks,

Author Response

Point 1: Thank you for the interesting paper. I have the following comments for the Authors.

Response 1: Thank you for your comments and suggestions that allowed us to greatly improve the quality of the manuscript. We agree with all your comments, and we corrected point by point the manuscript accordingly.

Point 2: Page1, Line 42:

The authors claim that are proposing “dual mode Fly-back converter”. However, this has been in application for long time (see J.N. Park - A DUAL MODE FORW ARD/FLYBACK

CONVERTER- IEEE Power Electronics Specialists conference 1982). Could you please explain the difference between the two concepts?

Response 2: To fully understand the paper “A DUAL MODEL FORWARD/FLYBACK CONVERTER”, it is important to present a single transistor dc-dc converter that combines both forward and fly-back action. There are mainly three kinds of advantage in the dual mode circuit. We used the following index numbers, as stated in the summary and conclusion section of this paper.

1). A transistor stress reduction due to supplying the transformer magnetizing energy to the load while effecting core reset.

2). For application wherein the load voltage is less than the peak value of the ac line, a large reduction in input filter capacitance may be realized due to the fly-back action.

3). A high ripple input operation significantly improves the ac mains power factor with only a small increase in the load filtering required.

And the mainly difference we proposing the dual-mode controller design for a quasi-resonant fly-back converter is that a dual mode control scheme combines the valley switching detection and fixed off-time control for improving the efficiency of a fly-back converter. In other word, the dual-mode control scheme not only enables a valley switching condition but also provides a constant frequency mechanism to reduce the conduction loss in QRF converter when operated in the continuous conduction mode and under heavy load condition. (Refer to Figure.10(b) and Figure 11(b) in a revised manuscript)

Point 3: Page 2, Line 55:

there is a typo in the “Conclusions are provided in section 6”. I think the authors mean to say section 5.

Response 3: We greatly appreciate your detailed and sorry for the mistake. We will recheck the sentence of draft for clarity in the revision of the manuscript. We decide to delete the paragraph in page 2 line 50-55 from the first manuscript. And then we add the following paragraph in the revised manuscript for reviewing the control method in literature [22] (This paragraph in page 2, line 46-52)

Point 4: Please use the equation numbers when substituting, or deriving one from the other: for example, in page 5 line 118, better to say “By substituting                                                into (16)” as you used correctly in line 120. Consistency is important for ease of reading.

Response 4: Thanks a lot for reading out work carefully and minutely as well as for giving us useful comments to improve its quality and readability. All the inconsistency sentence is corrected in blue word of a revised manuscript. We rearranged the sentence in the paragraph on page 5, line 123 of the revised manuscript. Please see the image below.

Point 5: Page 7, line 169: the passive components R, L, and are considered as having a crucial role in designing the parameters for QRF. It is important to give their specific importance in a sentence or two.

Response 5: We fully agree with the reviewer that the parameter of QRF is important in this proposed manuscript. We added the following sentence for addressing the reviewer’s question about the physical parameters in page 7, line176-177.

“For implementing these parameters, the R, L, and C were respectively 3.61Ω,170μH, and 2000μF at heavier load”.

Point 6: Figure 6 is too busy with unnecessary component values and the yellow color is also a wrong choice. You can list the values in a table for readability.

Response 6: According to reviewer’s suggestion, we will redraw the new Fig. 6 (Simplified circuit diagram of dual-mode control) in the revised manuscript for readability.

Point 7: Page 11, Line 232: “We are asked to increase …” I am not sure what you want to say here.

Response 7: We apologize for this grammar error in this paragraph, and we also thanks to reviewer for his patience on reading this paper.

For improving the English quality of the paper in order to improve its readability and writing structure, we rearranged the sentence in the paragraph on page 11, line 235-237 of the revised manuscript. We rewrite the following paragraph for clarifying the command.

“For fly-back converter stability, the phase margin should be positive. A literature book [21], for an unconditionally stable switch mode power conversion is to have a gain margin of 6dB and a phase margin of                                              

See [21], for more details.

[21] K. Kit Sum, Switch Mode Power Conversion, Marcel Dekker, New York, 1984.

We would like to sincerely thank you for your advices and constructive comments.

Sincerely,

Ching-Chun Chuang on behalf of all the authors.

Reviewer 3 Report

The last paragraph in the introduction is not necessary. There are some line numbers at the left had side of the paper which should be removed

It is not clear that how the small signal model has been derived using the method for PWM converters, Ref [18].

I could not find any information on the applications for this circuit

Author Response

Point 1: The last paragraph in the introduction is not necessary. There are some line numbers at the left had side of the paper which should be removed

Response 1: We agree with the command of the reviewer about the paragraph in the introduction. Both the last paragraph and line number (This paragraph in page 2, line 50-55) are removed from the draft manuscript. Moreover, we add the following paragraph in the revised manuscript for reviewing the control method in literature [22] (This paragraph in page 2, line 46-52)

Please see the image below

Point 2: It is not clear that how the small signal model has been derived using the method for PWM converters, Ref [18].

Response 2: Thanks a lot for taking your time and reading out work carefully as well as for giving us useful comments to improve its quality. We agree with the command of the reviewer about the literature book. We finally decided to use a literature book entitled “Fundamentals of Power Electronics” instead of the literature paper entitled “Effects of Parasitic Components on Diode Duty Cycle and Small-Signal Model of PWM DC-DC Buck Converter in DCM” for improving a readability. And also reschedule the literature in the revised manuscript. Please see the image below.

Point 3: I could not find any information on the applications for this circuit

Response 3: A dual-mode controller design for a quasi-resonant fly-back converter, which combines the valley-switching detection and fixed off-time control. (please see [4,20], for more details.). It mainly used for consumer electronics such as universal laptop adapter, and portable charger.

We would like to sincerely thank you for your advices and constructive comments.

Sincerely,

Ching-Chun Chuang on behalf of all the authors.

Round  2

Reviewer 1 Report

Authors have answered the reviewers' comments; however, there are still some English problems. Thanks.

Author Response

Response to Reviewer 1

The comments by the reviewer is really appreciated. Throughout, reviewer comments are in purple font and arial type, and our response in regular type. We have now revised the manuscript. In the following, we respond to the comments and the revised version of the manuscript will soon be transmitted.

Reviewer: Authors have answered the reviewers' comments; however, there are still some English problems. Thanks.

Authors: For improving the English quality of the paper in order to improve its readability and writing structure, we decide to use an expert reading proof serve available on internet for doing this task.

We apologize for all the grammar issue in the paper, and we also thanks to reviewers for his/her patient on reading this paper

We would like to sincerely thank you for your advices and constructive comments.

Sincerely,

Ching-Chun Chuang on behalf of all the authors.

Reviewer 2 Report

1) The use of the word "Initially" is confusing in the following sentence. (Is this to say in the previous works/publications??)

"Initially, the literature [10] proposed that the analysis of the active-clamp fly-back converter"

2) Introduction, Line 32

The following sentence needs attention. 

"In the concept of the literature paper, Fig. 2 and Fig. 3 show the seven"

Is this to say "to support/explain/illustrate the concept in the literature ..."?

3) Line 36, where do you show/calculate this percentage advantage or improvement in the paper?

"at least 4% in power stage efficiency... ". I can not locate this or give a reference where this 4% efficiency is shown.

4) "SPAACE" is not defined on page 2 line 47. What is it?

5) Page 11, Line 236, Better to say " As in [23, ]", "As shown in [23],"

6) Page 12, Line 255:  " The principle of a valley-switching detection technique in this research is:"

is this a section titler??

As a general note, I recommend the Authors to reread their manuscripts multiple times to make sure their idea is well articulated. 

Author Response

Response to Reviewer 2

The comments by the reviewer is really appreciated. Throughout, reviewer comments are in purple front and Arial type. We have now revised the manuscript. In the following, we respond to the comments and the revised version of the manuscript will soon be transmitted.

Reviewer: The use of the word "Initially" is confusing in the following sentence. (Is this to say in the previous works/publications??)

"Initially, the literature [10] proposed that the analysis of the active-clamp fly-back converter"

Author: we apologize for all the grammar issue in the paper, and we also thanks to reviewer for his/her patience on reading this paper. The word “Initially” is that in order to say that the previous work in literature [10]. For improving the paper readability and writing structure. We manually deleted a paragraph of the bundle we wanted to delete, which reveals a following paragraph to highlight the problems.          

Then, rewrite the newly sentence in the revised manuscript, as well as for clarifying the sentence of manuscript. See also paragraph of this section below. The newly paragraph will be updated in revised manuscript.

Reviewer: Introduction, Line 32

The following sentence needs attention. 

"In the concept of the literature paper, Fig. 2 and Fig. 3 show the seven"

Is this to say "to support/explain/illustrate the concept in the literature ..."?

Author: Yes. You are absolutely right. This mean that Fig.2 and Fig.2 illustrate the concept in the literature [10]. For improving the paper readability and writing structure. We rewrite the newly sentence in the revised manuscript, as well as for clarifying the sentence of manuscript. See also paragraph of this section below. The newly paragraph will be updated in revised manuscript.

Reviewer: Line 36, where do you show/calculate this percentage advantage or improvement in the paper? "at least 4% in power stage efficiency... ". I can not locate this or give a reference where this 4% efficiency is shown.

Author: literature [10] show that the efficiency comparison between RCD and active-clamp configuration in Fig. 4. It also said that the efficiency is improved 4% compared to the traditioned RCD configuration. For improving the paper readability and writing structure. The newly paragraph will be updated in revised manuscript.

Reviewer: "SPAACE" is not defined on page 2 line 47. What is it

Author: Thanks a lot for reading out work carefully and minutely as well as for giving us useful comments to improve its quality and readability. For improving the paper readability and writing structure, we defined the SPAACE in the page 1-2 line 44-45 in the revised manuscript. Please see below image.

Reviewer: Page 11, Line 236, Better to say " As in [23, ]", "As shown in [23],"

Author: Thanks a lot for reading out work carefully. We rewrite the sentence in the page 11 line 232 in the revised manuscript. Please see below image.

Reviewer: Page 12, Line 255:  " The principle of a valley-switching detection technique in this research is:" is this a section titler??

Author: The sentence The principle of a valley-switching detection technique in this research is:” is not a section titler. We apologize for all the grammar issue in the paper. We rearrange the sentence in the page 12 line 250-251. Please see below image.

Reviewer: As a general note, I recommend the Authors to reread their manuscripts multiple times to make sure their ideal is well articulated.

Author: Thank you for your comments and suggestions that allowed us to greatly improve the quality of the manuscript.

 We would like to sincerely thank you for your advices and constructive comments.

Sincerely,

Ching-Chun Chuang on behalf of all the authors.

Reviewer 3 Report

No comments.

Author Response

Response to Reviewer 3

The comments by the reviewer is really appreciated. Throughout, reviewer comments are in purple front and Arial type. We have now revised the manuscript. In the following, we respond to the comments and the revised version of the manuscript will soon be transmitted.

Reviewer: No comments

Authors: We expect and appreciate the effort of the reviewers to the future.

We would like to sincerely thank you for your advices and constructive comments.

Sincerely,

Ching-Chun Chuang on behalf of all the authors.
